# Clinical Prognosis of Lung Cancer in Patients with Moderate Chronic Kidney Disease

**DOI:** 10.3390/cancers14194786

**Published:** 2022-09-30

**Authors:** Taehee Kim, Sang Hyuk Kim, Hayoung Choi, Tae Rim Shin, Hwan Il Kim, Seung Hun Jang, Ji Young Hong, Myung Goo Lee, Soojie Chung, In Gyu Hyun, Yun Su Sim

**Affiliations:** 1Division of Pulmonary, Allergy and Critical Care Medicine, Department of Internal Medicine, Hallym University Kangnam Sacred Heart Hospital, Seoul 07441, Korea; 2Lung Research Institute, Hallym University College of Medicine, Chuncheon 24252, Korea; 3Division of Pulmonary, Allergy and Critical Care Medicine, Department of Internal Medicine, Hallym University Sacred Heart Hospital, Anyang-si 14068, Korea; 4Division of Pulmonary, Allergy and Critical Care Medicine, Department of Internal Medicine, Hallym University Chuncheon Sacred Heart Hospital, Chuncheon 24252, Korea; 5Division of Pulmonary, Allergy and Critical Care Medicine, Department of Internal Medicine, Hallym University Dongtan Sacred Heart Hospital, Dongtan-si 18450, Korea

**Keywords:** lung cancer, chronic kidney disease, mortality

## Abstract

**Simple Summary:**

Lung cancer has a high incidence and mortality rate worldwide. In addition, lung cancer develops commonly in the elderly, and it is necessary to consider comorbidities when planning treatment. Chronic kidney disease (CKD) is a common comorbidity in patients with lung cancer. However, there are conflicting results regarding its effect on the clinical prognosis of lung cancer, and only insufficient evidence for treatment of lung cancer according to renal function. In this retrospective multicenter study, we evaluate clinical course and prognostic factors of lung cancer according to the renal function of moderate CKD patients.

**Abstract:**

The clinical outcomes of patients with lung cancer coexisting with chronic kidney disease (CKD) are reported to have been conflicting. There is insufficient evidence for treatment and prognosis of lung cancer according to renal function in patients with CKD. We evaluate clinical course and prognostic factors of lung cancer according to the renal function of moderate CKD patients. A retrospective, multicenter study of lung cancer patients with moderate CKD was performed. Moderate CKD was defined as having an estimated glomerular filtration rate (eGFR) < 60 mL/min/1.73 m^2^. CKD was classified as stage 3, stage 4, and stage 5 according to eGFR. The cumulative mortality of lung cancer was calculated by competing risks survival analysis, and the risk factors were evaluated by the Cox-proportional hazards model. Among the lung cancer patients with moderate CKD (n = 181), median overall survival (OS) was 11.1 (4.2–31.3) months for stage 3 CKD patients, 6.0 (1.8–16.3) months for stage 4 CKD patients, and 4.7 (2.1–40.1) months for stage 5 CKD patients (*p* = 0.060), respectively. In a subgroup analysis, CKD stage was associated with an increased mortality in early-stage non-small cell lung cancer (NSCLC). Cox regression analysis revealed that age ≥ 75 years (adjusted hazard ratio (aHR), 1.581; 95% confidence interval (CI), 1.082–2.310), Charlson comorbidity index (aHR, 1.669; 95% CI, 10.69–2.605), and stage IV NSCLC (aHR, 2.395; 95% CI, 1.512–3.796) were associated with increased mortality risk, whereas adenocarcinoma (aHR, 0.580; 95% CI, 0.352–0.956) and stage 3 CKD (aHR, 0.598; 95% CI, 0.399–0.895) were associated with decreased mortality risk. In conclusion, the mortality risk of patients with lung cancer was lower in stage 3 CKD compared with stage 4 or 5 CKD. In addition, in the early stages of NSCLC, the CKD stage affected the prognosis, but not in the advanced stage NSCLC.

## 1. Introduction

Lung cancer is the leading cause of cancer-related mortality worldwide [1]. The comorbidities at the time of lung cancer diagnosis are one of the important factors that affect prognosis [2]. Lung cancer is associated with advanced age and smoking, both of which are also associated with comorbidities [3,4]. Lung cancer is particularly common in the elderly, and the average age at diagnosis is 70 years [5]. Chronic kidney disease (CKD) is also common in the elderly due to an approximately 1% per year decrease in the average creatinine clearance (CrCl) [6].

CKD is associated with poor prognosis in various type of cancer compared to non-CKD patients [7,8]. The impact of CKD on the prognosis of lung cancer patients remains controversial. Several studies reported that the presence or absence of CKD in lung cancer patients does not affect the prognosis of lung cancer [9,10]. By contrast, in a large population-based cohort study, Wei et al. [11] found an increased mortality rate in lung cancer patients with CKD compared to those without CKD. Therefore, the outcome of lung cancer patients with CKD remains unclear. Furthermore, there is insufficient evidence for lung cancer management according to the CKD stage (i.e., degree of renal impairment). The purpose of the present study was to analyze the clinical course and prognostic factors of lung cancer according to the renal function of moderate CKD patients, including pre-dialysis and hemodialysis patients.

## 2. Methods

### 2.1. Study Design

The study enrolled 5351 patients pathologically diagnosed with lung cancer between 1 January 2008 and 31 December 2019 at four university teaching hospitals. We retrospectively analyzed the medical records of 190 patients with stage 3 or higher CKD. Nine patients for whom treatment was discontinued due to transfer to another hospital were excluded.

### 2.2. Diagnosis of Lung Cancer

Lung cancer was pathologically confirmed on the basis of malignant cells in biopsy specimens, pleural fluid, or bronchial lavage fluid. Staging of lung cancer was assessed according to the *International Association for the Study of Lung Cancer Staging Manual in Thoracic Oncology*, 8th edition [12].

### 2.3. Definition of Moderate Chronic Kidney Disease

Renal function was evaluated as the estimated glomerular filtration rate (eGFR), calculated as: eGFR (mL/min/1.73 m^2^) = 194 × serum creatinine^−1.094^ (mg/dL) × age^−0.278^ (if female, 194 was replaced with 90.739). CKD was defined as an eGFR <90 mL/min/1.73 m^2^ with (1) a history of CKD, (2) follow-up at the hospital for more than 3 months, or (3) morphological change was confirmed at the time of lung cancer diagnosis [13]. CKD was classified as stage 3 based on an eGFR of 30–60 mL/min/1.73 m^2^, stage 4 based on an eGFR of 15–30 mL/min/1.73 m^2^, and stage 5 based on an eGFR of <15 mL/min/1.73 m^2^ or need of dialysis. All stage 5 CKD patients enrolled in this study were on hemodialysis.

### 2.4. Variables

We assessed the demographic and clinicopathological information of the patients, including their age; sex; body mass index (BMI); epidermal growth factor receptor (EGFR), anaplastic lymphoma receptor tyrosine kinase, and programmed cell death protein-ligand 1 (PD-L1), treatment regimen; and progression-free survival (PFS), overall survival (OS), and 5-year survival rate. The evaluation of the treatment response for lung cancer was confirmed by the clinician’s medical record based on the chest CT results performed every 1–3 months. PFS was evaluated only in the first chemotherapy and was defined as the time from first chemotherapy commencement to either documented disease progression or death from any cause. OS was defined as the time from pathological diagnosis to death from any cause. Progression was evaluated on the basis of response using Response Evaluation Criteria in Solid Tumors (version 1.1) [14].

### 2.5. Statistical Analysis

Frequencies are expressed as numbers (%) and descriptive data are expressed as medians with interquartile range (IQR). We compared the clinical features, pathological type of lung cancer, stage, treatment regimen, OS, PFS, and 5-year survival according to CKD groups. The Chi-square test or Fisher’s exact test was used to analyze categorical variables, and the Kruskal–Wallis test was used to analyze continuous variables. Cumulative incidence curves were used in a competing risk setting to calculate the probability of high risk for renal dysfunction in CKD patients. Gray’s test was used to compare cause-specific cumulative incidence curve. The association of CKD with lung cancer outcomes were evaluated using Cox proportional-hazards model. Univariate analysis was performed to identify prognostic factors related to OS by creating a categorical variable based on the median values of the clinical characteristics, pathological type, lung cancer stage, and CKD groups at the time of lung cancer diagnosis in CKD patients. Factors significantly associated with OS were further analyzed in a Cox proportional hazard model to adjust for the potential confounding effects of each factor. Hazard ratios (HR) with 95% confidence intervals (CI) were calculated. The seven variables included in the Cox regression analysis were sex and the factors with a *p* value < 0.05 in the univariate analysis.

## 3. Results

### 3.1. Clinical Characteristics of Patients with Lung Cancer in CKD

Table 1 presents the clinical characteristics of the enrolled patients. The patients had a median age of 75 (70–81) years, and 87% (157/181) of the patients were males. In patients with lung cancer in CKD, the median creatinine level was 1.68 (1.45–2.29) mg/dL and the CrCl was 35.9 (23.8–45.0) mL/min/1.73 m^2^. Comorbidities in lung cancer with CKD patients were hypertension, 68% (123/181); diabetes mellitus, 47% (85/181); chronic airway disease, 28% (50/181); heart disease, 18% (33/181); other cancer, 18% (33/181); old cerebral disease, 12% (22/181). The pathological type of lung cancer was non-small cell lung cancer (NSCLC) in 83% (150/181) and small cell lung cancer (SCLC) in 17% (31/181) of the patients. Among the NSCLC patients, adenocarcinoma was diagnosed in 41% (61/181), squamous cell carcinoma in 45% (68/181), and large cell carcinoma in 5% (8/181) of the patients. Thus, squamous cell carcinoma was the most common pathological type in enrolled patients. TNM stages of patients diagnosed with NSCLC, stage I–IV cancers were observed in 23% (35/150), 11% (16/150), 23% (34/150), and 43% (65/150), respectively. Among the SCLC patients, 26% (8/31) and 74% (23/31) had limited and extensive disease, respectively. The lung cancer patients with CKD were treated with surgery (25%; 45/181), radiotherapy (25%; 45/181), first-line chemotherapy (37%; 67/181), combined chemotherapy and radiotherapy (10%; 18/181), targeted therapy (12%; 21/181), or palliative supportive therapy (31%; 56/181). Selected treatment options for lung cancer were shown in Figure 1.

### 3.2. Clinical Outcomes in Patients with Lung Cancer According to CKD Stage

During the follow-up period, there were 151 (83%) deaths in patients with lung cancer in CKD. According to the CKD stage, 80% (93/117) of patients with stage 3 CKD, 90% (38/42) with stage 4 CKD, and 90% (20/22) with stage 5 CKD (*p* = 0.156). The median PFS was 4.9 (2.7–106) months for all CKD patients, 4.7 (2.9–7.8) months for stage 3 CKD patients, 4.9 (0.8–10.6) months for stage 4 CKD patients, and 7.2 (2.7–18.2) months for stage 5 CKD patients (*p* = 0.456). The median OS was 10.2 (3.3–28.3) months for all CKD patients, 11.1 (4.2–31.3) months for stage 3 CKD patients, 6.0 (1.8–16.3) months for stage 4 CKD patients, and 4.7 (2.1–40.1) months for stage 5 CKD patients (*p* = 0.060). Detailed information of clinical outcomes presented in Table 2.

To evaluate the effect of competitive risk factors on mortality from lung cancer, a subgroup analysis was performed in patients with NSCLC. The results of competing risks survival analysis according to CKD stage in early NSCLC (stage I–IIIA, Figure 2A) and advanced NSCLC (stage IIIB–IV, Figure 2B) are presented. The effect of CKD stage on death in early NSCLC (stage ≤ IIIA) was significantly highest in the order of CKD4, CKD 5 (hemodialysis), and CKD 3 (Gray’s test, *p* < 0.001).

### 3.3. Clinical Outcomes of Patients with Lung Cancer in CKD

Table 2 presents the clinical outcomes of lung cancer patients with CKD. Among all CKD patients with lung cancer, the 5-year survival rate was 17% for patients aged less than 75 years and 3% for those aged more than 75 years. The 5-year survival rate was 11% for NSCLC patients (21% for adenocarcinoma and 6% for squamous cell carcinoma) and 0% for SCLC patients. In the survival outcome of NSCLC by TNM stage, the 5-year survival rate was 34% for stage I, 0% for stage II, 3% for stage III, 3% for stage IV. The OS was 51.9 (20.0–71.5) months in stage I, 14.5 (5.0–40.0) months in stage II, 11.1 (4.4–31.3) months in stage III, and 6.6 (3.1–14.4) months in stage IV. Among SCLC patients, the OS was 4.1 (1.5–21.4) months with limited disease and 2.1 (0.9–8.2) months with extensive disease. Clinical outcomes according to treatment options for lung cancer are presented in Appendix A.

### 3.4. Prognostic Factors of Patients with Lung Cancer in CKD

Univariate and multivariate Cox regression analyses were performed to determine the prognostic factors for lung cancer patients with CKD (Figure 3). From the univariate analysis, old age (≥75 years), a low BMI (≤23 kg/m^2^), Charlson comorbidities index (CCI ≥ 10), comorbid chronic airway disease, SCLC, stage IV NSCLC, absolute monocyte count (AMC ≥ 500, /μL), and neutrophil–lymphocyte ratio (NLR ≥ 3.37) were associated with poor prognosis in terms of reduced OS (all *p* < 0.05). From the multivariate analysis, age ≥ 75 years (aHR, 1.658; 95% CI, 1.148–2.394, *p* = 0.007), CCI ≥ 10 (aHR, 1.669; 95% CI, 1.069–2.605, *p* = 0.024) and stage IV NSCLC (aHR, 2.395; 95% CI, 1.512–3.796, *p* < 0.001) were independent prognostic factors. By contrast, the pathological type of adenocarcinoma (aHR, 0.580; 95% CI, 0.352–0.956, *p* = 0.033) and stage 3 CKD (aHR, 0.598; 95% CI, 0.399–0.895, *p* = 0.012) were associated with reduced mortality risk. Figure 4 shows the Cox survival curves for age ≥ 75 years (Figure 4A), CCI ≥ 10 (Figure 4B), adenocarcinoma (Figure 4C), and CKD stage 3 (Figure 4D), respectively.

## 4. Discussion

In the present study, moderate CKD was present in 3.6% of newly diagnosed lung cancer patients. A previous cohort study conducted in Taiwan reported an incidence of 1.7% for lung cancer coexisting with CKD [11]. However, the previous study defined CKD on the basis of International Classification of Disease codes from health insurance data; thus, the incidence of moderate CKD was higher than that in our study, which defined CKD on the basis of the eGFR.

Lung cancer is histopathologically classified as NSCLC or SCLC. NSCLC accounts for more than 80–85% of lung cancers, of which almost 40% are adenocarcinomas and 25–30% are squamous cell carcinomas [15,16,17]. In the present study, the histological type of lung cancer was SCLC in 17% and NSCLC in 83% of the patients, similar to the general proportions based on lung cancer histology. Among the NSCLC patients, squamous cell carcinoma was the most common type (45%), followed by adenocarcinoma (41%). In a retrospective study of 671 patients who underwent pulmonary resection for NSCLC, Yamamoto et al. [18] reported that squamous cell carcinoma was more frequently diagnosed in CKD patients than in non-CKD patients (34.5% vs. 15.4%, respectively; *p* < 0.01). Cigarette smoking is a stronger risk factor for squamous cell carcinoma than for adenocarcinoma, and the proportion of smoking was higher in CKD patients than in non-CKD patients in the previous study.

In the present study, we analyzed the clinical course of lung cancer in moderate CKD patients and found that old age (≥75 years), CCI (≥10), and stage IV NSCLC were poor prognostic factors, whereas adenocarcinoma and stage 3 CKD were good prognostic factors. The older patients (≥75 years) had a 1.58-fold higher mortality rate than the younger patients. Mortality due to lung cancer generally peaks at 85 years, consistent with our results [19]. As a result of subgroup analysis, although there was no difference in the stage of lung cancer, surgery (38.8%) was the most frequently preferred treatment option for patients aged < 75 years, whereas only supportive treatment (44.8%) was preferred for patients aged ≥ 75 years (*p* < 0.001).

The median CCI value of the subjects in this study was 10.0 (8.0–11.0), and patients with CCI higher than this had an approximately 1.7-times higher risk of mortality. Marcus et al. [20] shown that higher CCI was associated with higher lung cancer-specific mortality (CCI ≥ 3: aHR, 5.16; 95% CI, 2.07–12.89). Lu et al. [10] reported that median CCI was 8.0 in moderate CKD patients with lung cancer, and median survival for patients with a CCI > 9 was 6 months compared with 10 months in patients with CCI < 9. Comorbidities have a significant impact on the survival of lung cancer patients, and the number and severity of comorbidities are likely to influence treatment options for these patients.

In addition, stage IV NSCLC was an independent poor prognostic factor for lung cancer patients with coexistent CKD. In the present study, the 5-year survival rates for NSCLC were 34% for stage I patients, 0% for stage II patients, 9% for stage III patients, and 3% for stage IV patients. The 5-year survival rates were lower in the present study than in a previous study that reported 5-year survival rates of 68–92% for stage I NSCLC, 53–60% for stage II NSCLC, 13–36% for stage III NSCLC, and 0–10% for stage IV NSCLC [21]. The subgroup analysis showed that 36% (65/181) of the patients had stage IV NSCLC at diagnosis and received palliative chemotherapy (49.2%) or supportive treatment (38.5%). In a large retrospective study of medical cost according to the treatment modality and disease stage, Cipriano et al. [22] reported that 13.9% of stage IV NSCLC patients did not receive anti-cancer treatment. Similarly, in the present study, anti-cancer treatment was not administered to a large proportion of stage IV NSCLC patients with CKD, which may explain the poor prognosis of these patients.

By contrast, adenocarcinoma pathology and stage 3 CKD were favorable prognostic factors in this study. *EGFR* mutations are the most common target driver mutations found in lung adenocarcinoma and are detected in 62% of the Asian population [23,24]. In lung adenocarcinoma, the median OS was 14 months when treated with pemetrexed-platinum doublet chemotherapy, but the value increased to 39 months upon treatment with a third-generation *EGFR* tyrosine kinase inhibitor (TKI) [25,26,27]. In the present study, 61 patients were diagnosed with lung adenocarcinoma, of whom 16 had *EGFR* mutations and 15 were treated with EGFR-TKIs. The survival analysis, although not reached statistically significant, revealed a longer median OS in the EGFR-TKI treatment group compared to the non-EGFR-TKI treatment group (21.7 vs. 15.8 months; *p* = 0.593).

According to the renal function, the mortality risk of patients with lung cancer was reduced by 40% in CKD stage 3 compared CKD 4 or 5 stages in this study. Selected treatment options for lung cancer according to CKD stage in early and advanced lung cancer stage are presented in Appendix A. The rate of early diagnosis (i.e., at cancer stages I–IIIA) was highest in stage 5 CKD patients (40.9%) compared to stage 3 and 4 patients (37.6% and 40.5%, respectively), which may be explained by the frequent chest X-rays being performed on stage 5 CKD patients who are receiving hemodialysis. The proportions of patients who underwent surgical treatment in patients with resectable lung cancer stage I–IIIA were 63.6%, 47.1%, and 66.7% for stage 3–5 CKD, respectively. However, in stage I–IIIA lung cancer patients, the 5-year survival rates were 25.0% for stage 3 CKD, 5.9% for stage 4 CKD, and 22.2% for stage 5 CKD. Previous studies have reported 5-year survival rates of 28–43% after lung resection in hemodialysis patients with lung cancer, which was non-inferior to those among patients not receiving hemodialysis [18,28]. In this study, the survival rate was highest among patients with stage 3 CKD, probably because they frequently received adjuvant chemotherapy after lung resection. Almost 84.6%, 15.4%, and 0.0% of patients with stage 3–5 CKD received adjuvant chemotherapy, respectively. In other words, the early diagnosis rate was higher in stage 5 CKD patients compared to stage 3 CKD patients, whereas the rate of surgery was similar. Nevertheless, the poor survival rate for stage 5 CKD patients was likely due to the low adjuvant chemotherapy rate.

Lung cancer appears to be less affected by the presence or absence of CKD than other types of cancer. Weng et al. [7] studied whether CKD acts as a poor prognostic factor in various types of cancer. As a result, the presence of CKD adversely affected liver cancer, kidney and urinary tract cancer, but not lung cancer. Na et al. [8] also compared the prevalence and prognosis of various cancer types according to the presence or absence of CKD, the presence of CKD was an independent poor prognostic factor in cancer-specific mortality except for lung cancer. They explained that due to the relatively rapid progression of lung cancer, it has shown a similar clinical course and survival compared to lung cancer patients than those without CKD. In a study conducted by Lu et al. [10] of lung cancer patients with moderate CKD, there is no effect on mortality according to lung cancer stage or CKD stage. In this study, the degree of renal impairment influenced mortality but not progression in early-stage NSCLC (I-IIIA). However, the stage of CKD did not affect both progression and mortality in advanced stage NSCLC (IIIB-IV). In other words, although renal function affects the clinical course of early NSCLC, the aggressiveness of lung cancer itself seems to impact more on the prognosis in advanced NSCLC.

Although the study results are limited by the retrospective study design and relatively small sample size, the clinical outcomes of lung cancer in patients with CKD were significantly influenced by factors related to renal function as well as lung cancer characteristics (old age, cancer stage, and pathological type). Additionally, despite patients with stage 3 CKD having a better lung cancer prognosis than stage 4 or 5 CKD patients, the former received more aggressive lung cancer treatment than patients with impending or definitive end-stage renal disease (i.e., stage 4 or 5 CKD, respectively).

## 5. Conclusions

The mortality risk of patients with lung cancer was lower in stage 3 CKD compared stage 4 or 5 CKD. In addition, in the early stage of NSCLC, mortality rate was affected according to the CKD stage, but not in the advanced stage NSCLC. Although the treatment of lung cancer should be individualized according to renal function, limited information is available. Therefore, further large-scale studies are required to determine the optimal treatment for lung cancer in patients with renal dysfunction.

## Figures and Tables

**Figure 1 cancers-14-04786-f001:**
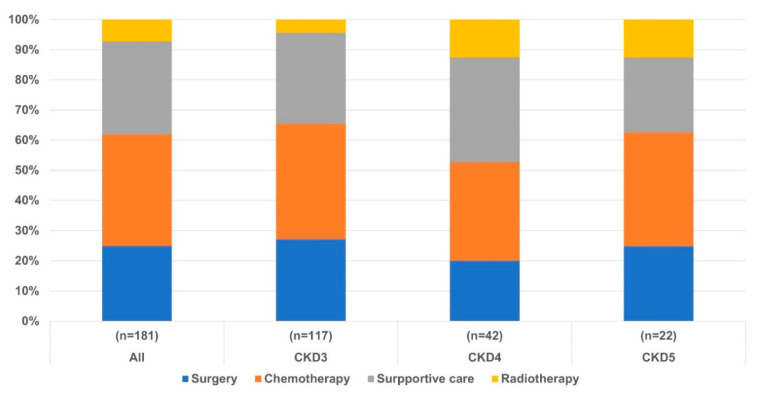
Selected treatment options for lung cancer according to CKD stage in early and advanced lung cancer stage.

**Figure 2 cancers-14-04786-f002:**
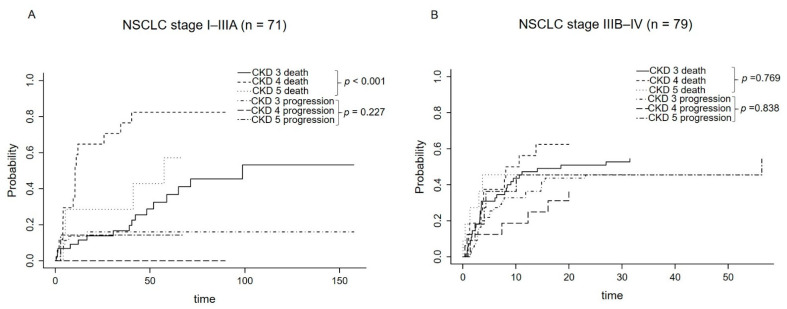
Cumulative incidence rate for progression and death from NSCLC in competing risks survival analysis stratified by the stage of CKD. (**A**) early NSCLC (stage ≤ IIIA), (**B**) advanced NSCLC (stage ≥ IIIB). Cumulative incidence rate for death from early NSCLC (stage ≤ IIIA) was lower in CKD stage 3.

**Figure 3 cancers-14-04786-f003:**
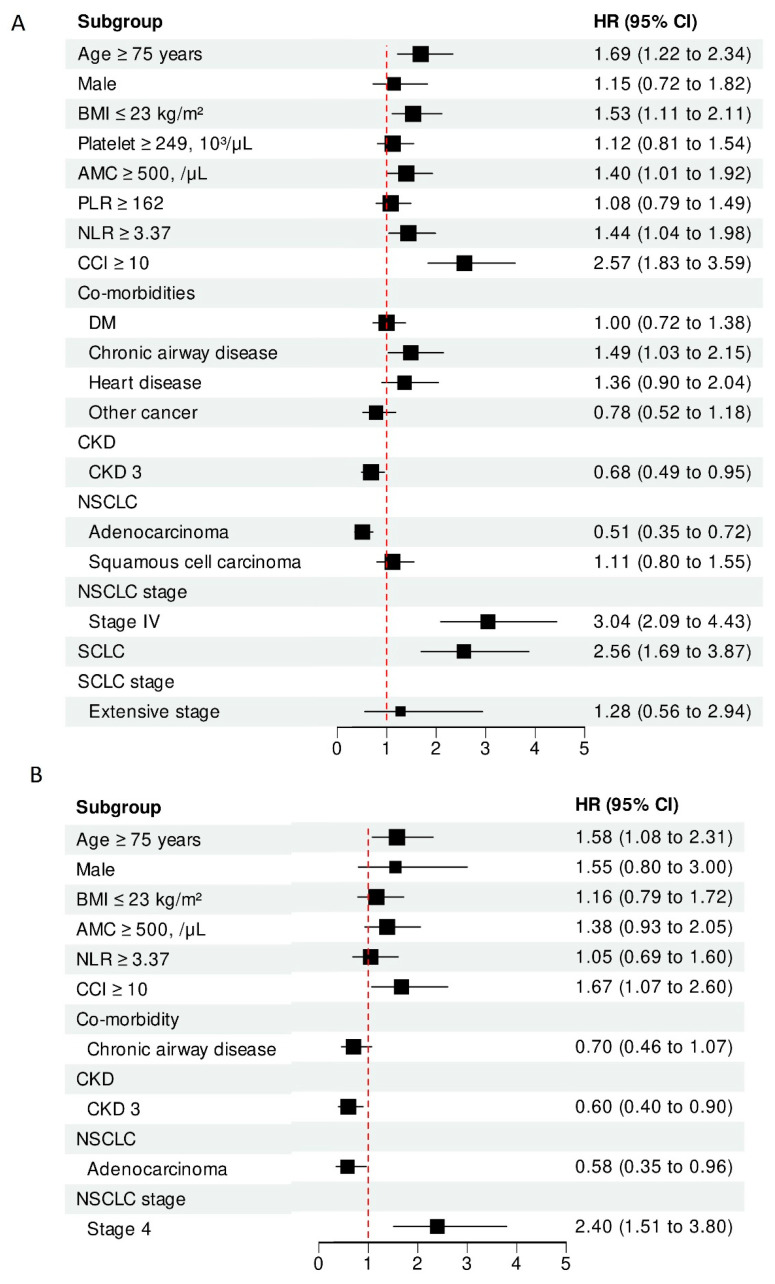
A forest plot of (**A**) univariate regression analysis and (**B**) multivariate regression analysis.

**Figure 4 cancers-14-04786-f004:**
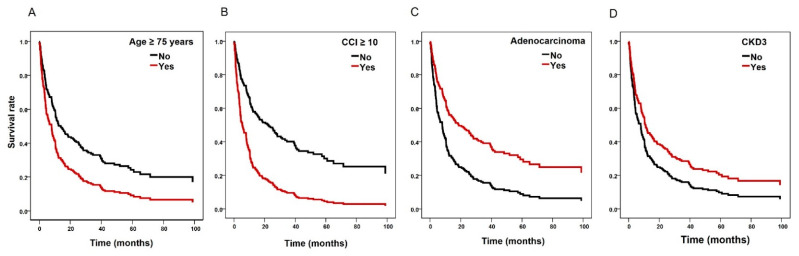
Cox’s proportional hazards regression model for overall survival of lung cancer patients with moderate chronic kidney disease stage (CKD). (**A**) old age (≥75 years), (**B**) Charlson comorbidities index (≥10), (**C**) pathologic type of adenocarcinoma, and (**D**) CKD stage 3.

**Table 1 cancers-14-04786-t001:** Clinical characteristics and treatment of lung cancer patients with chronic kidney disease.

Characteristics	All Patients (n = 181)	CKD 3 (n = 117)	CKD 4 (n = 42)	CKD 5 (n = 22)	*p*-Value
Age, year	75 (70–81)	75 (71–81)	79 (71–82)	73 (64–78)	0.040
Male	157 (87%)	104 (89%)	35 (83%)	18 (82%)	0.507
Body mass index, kg/m^2^	23.0 (21.4–25.1)	23.4 (22.0–25.7)	22.0 (19.8–24.9)	22.0 (20.0–24.0)	0.007
BUN, mg/dL	24.6 (19.1–31.8)	22.0 (17.5–25.6)	32.0 (28.0–45.4)	44.6 (25.9–54.1)	<0.001
Creatine, mg/dL	1.68 (01.45–2.29)	1.5 (1.4–1.68)	2.4 (1.8–3.2)	6.3 (5.0–8.3)	<0.001
Creatine clearance, mL/minute/1.73 m^2^	35.9 (23.8–45.0)	40.9 (36–48)	23.0 (17.3–26.4)	8.9 (6.8–8.3)	<0.001
Charlson comorbidities index	10.0 (8.0–11.0)	10.0 (8.0–11.0)	10.0 (7.8–11.0)	10.0 (6.8–11.3)	0.920
Comorbidity					
Hypertension	123 (68%)	82 (70%)	29 (69%)	12 (55%)	0.353
Diabetes	85 (47%)	50 (43%)	22 (52%)	13 (59%)	0.268
Chronic airway disease	50 (28%)	33 (28%)	13 (31%)	4 (18%)	0.540
Heart disease	33 (18%)	17 (15%)	9 (21%)	7(32%)	0.130
Other cancer	33 (18%)	22 (19%)	8 (19%)	3 (14%)	0.837
Old cerebral disease	22 (12%)	12 (12%)	5 (12%)	3 (14%)	0.975
Pathologic type					
NSCLC	150 (83%)	99 (85%)	33 (79%)	18 (82%)	0.665
Adenocarcinoma	61 (41%)	38 (38%)	12 (36%)	11 (61%)	0.203
Squamous cell carcinoma	68 (45%)	50 (51%)	13 (39%)	5 (28%)	0.124
Large cell carcinoma	8 (5%)	5 (5.1%)	3 (10%)	0	0.415
Sarcoma	2 (1%)	1 (1.0%)	1 (3.0%)	0	0.625
Undifferentiated NSCLC	11 (7%)	5 (5.1%)	4 (12%)	2 (11%)	0.388
SCLC	31 (17%)	18 (15%)	9 (21%)	4 (18%)	0.665
EGFR (n = 107)	18 (17%)	10 (9%)	4 (4%)	4 (4%)	0.387
ALK (n = 54)	1 (2%)	0	0	1 (1.9%)	<0.001
PD-1 (n = 49)	26 (53%)	20(41%)	3 (6%)	3 (6%)	0.017
NSCLC stage					
Stage I	35 (23%)	21 (21%)	7 (21%)	7 (39%)	0.251
Stage II	16 (11%)	10 (10%)	6 (18%)	0	0.126
Stage III	34 (23%)	23 (23%)	6 (18%)	5 (28%)	0.717
Stage IV	65 (43%)	45 (46%)	14 (42%)	6 (33%)	0.630
SCLC stage					
Limited stage	8 (26%)	4 (22%)	1 (11%)	3 (75%)	0.045
Extensive stage	23 (74%)	14 (78%)	8 (89%)	1 (25%)	0.045
Complete blood cell count					
Platelet, 10³/μL	249 (194–302)	260 (207–310)	230 (176–287)	187 (146–260)	0.001
AMC, /μL	500 (400–649)	500 (400–640)	523(395–677)	463 (358–660)	0.595
PLR	162 (117–243)	171 (118–246)	176 (121–271)	117 (90–193)	0.086
NLR	3.37 (2.45–5.10)	3.37 (2.46–4.76)	3.71 (2.29–6.37)	3.19 (2.65–5.31)	0.677

Data are presented as the median value (interquartile range) or number. CKD, chronic kidney disease; NSCLC, non-small cell lung cancer; SCLC, small cell lung cancer; EGFR, epidermal growth factor receptor; ALK, anaplastic lymphoma receptor tyrosine kinase; PD-1, programmed cell death protein-1; AMC, absolute monocyte count; NLR, neutrophil-lymphocyte ratio; PLR, platelet–lymphocyte ratio.

**Table 2 cancers-14-04786-t002:** Clinical outcome of lung cancer patients with chronic kidney disease.

		All Patients		CKD 3		CKD 4		CKD 5		
		(n = 181)		(n = 117)		(n = 42)		(n = 22)		*p*-Value
PFS (months)		4.9 (2.7–106)		4.7 (2.9–7.8)		4.9 (0.8–10.6)		7.2 (2.7–18.2)		0.456
Death		151 (83%)		93 (80%)		38 (90%)		20 (90%)		0.156
OS (months)	5year survival	10.2 (3.3–28.3)	17 (9%)	11.1 (4.2–31.3)	12 (10%)	6.0 (1.8–16.3)	2 (5%)	4.7 (2.1–40.1)	3 (14%)	
Age ≥ 75 years		6.1 (2.1–18.5)	3 (3%)	6.5 (2.4–18.6)	2 (3.7%)	4.1 (1.8–14.2)	0	4.5 (2.1–57.4)	1 (14%)	0.694
< 75 years		13.8 (4.7–40.8)	14 (17%)	22.6 (8.5–50.0)	12 (19%)	10.0 (1.6–22.8)	2 (13%)	4.8 (1.8–39.7)	2 (13%)	0.044
Pathologic type										
NSCLC		11.1 (4.0–35.0)	17 (11%)	15.6 (5.7–39.5)	12 (12%)	9.5 (3.6–25.6)	2 (6%)	4.7 (2.7–45.2)	3 (16%)	0.122
ADC		21.7 (8.1–50.8)	13 (21%)	17.7 (9.8–53.5)	8 (21%)	31.4 (8.6–57.9)	2 (17%)	11.5 (4.5–60.9)	3 (27%)	0.686
Sqcc		8.8 (3.3–26.2)	4 (6%)	12.6 (4.2–28.8)	4 (8%)	4.0 (1.6–9.5)	0	3.7 (0.2–22.7)	0	0.001
SCLC		3.4 (1.0–8.4)	0	4.4 (1.1–8.5)	0	1.7 (0.9–3.6)	0	3.0 (1.5–33.1)	0	0.194
NSCLC stage										
Stage I		51.9 (20.0–71.5)	12 (34%)	53.5 (29.2–98.5)	9 (43%)	34.5 (10.4–58.9)	1 (14%)	41.2 (5.3–62.2)	2 (29%)	0.186
Stage II		14.5 (5.0–40.0)	0	22.3 (6.6–46.3)	0	10.0 (3.3–29.4)	0	-	-	0.151
Stage III		11.1 (4.4–31.3)	3 (9%)	20.6 (9.1–50.0)	3 (13%)	6.1 (1.8–10.6)	0	3.7 (0.7–18.9)	0	0.002
Stage IV		6.6 (30.1–14.4)	2 (3%)	8.2 (3.3–15.1)	0	5.9 (2.1–16.7)	1 (7%)	3.8 (1.5–18.8)	1 (17%)	0.875
SCLC stage										
Limited stage		4.1 (1.5–21.4)	0	4.1 (1.2–5.5)	0	3.23	0	25.6 (0.9–39.7)	0	0.295
Extensive stage		2.1 (0.9–8.2)	0	4.4 (1.1–9.3)	0	1.4 (0.9–3.5)	0	2.13	0	0.421

Data are presented as the median value (interquartile range) or number. *p*-values were obtained using Cox survival analysis, CKD, chronic kidney disease; NSCLC, non-small cell lung cancer; ADC, adenocarcinoma; Sqcc, Squamous cell carcinoma; SCLC, small cell lung cancer.

## Data Availability

The data presented in this study are available on request from the corresponding author.

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
