# Peer review of "Clinical Prognosis of Lung Cancer in Patients with Moderate Chronic Kidney Disease"

_cancers, 2022, doi:10.3390/cancers14194786_

Round 1
Reviewer 1 Report
Journal: Cancers
Manuscript Title: Clinical prognosis of lung cancer in patients with moderate chronic kidney disease
Manuscript Number: cancers-1912322
Reviewer Comments:
Although this is a good research paper on Clinical prognosis of lung cancer patients with chronic kidney disease, there are several major points that should be addressed:
1) The authors claim that they used Chest computed tomography (CT), however there are no data in the results. The authors should add these CT images in order to evaluate the treatment response.
2) The authors should also analyze CKD prevalence and absolute number of CKD patients in different cancer types. They should also perform Multiple logistic regression of various cancer types for CKD.
3) Circulating blood counts of ALC, AEC1, AMC, Platelet count, NLR1 PLR1 etc should be included in the results analysis
4) Clinical analysis of patients should include a detailed cytokine analysis of TNF-a, IL-6, IL8, IP-10, IL-6, CXCL5, CXCL9, and CXCL10, CXCL9, CXCL10, CXCL11, CXCL19, etc. The authors should also compare cytokine levels before and after treatment.
5) The authors have no data of Survival according to CKD during chemotherapy. They should add Survival data on Time to treatment failure (TTF), progression-free survival (PFS), and overall survival (OS) of patients with CKD during chemotherapy.
6) The authors should also add Kaplan–Meier survival curve for lung cancer patients with and without chronic kidney disease (CKD) for comparison.
7) The authors do not display any data of Cancer stage survival according to the presence or not of chronic kidney disease and stage of chronic kidney disease. It is really important in order to verify their results
8) The authors should add a distribution analysis of Charlson comorbidities index (CCI) in CKD and non-CKD lung cancer patients
Overall this paper is presented with low scientific novelty and originality and cannot be accepted in its current form. Major revision should be performed.
Author Response
Point-by-point response to the reviewers’ comments
Manuscript title: Clinical prognosis of lung cancer in patients with moderate chronic kidney disease
To the reviewers
We sincerely appreciate the time and effort you invested to help us improve this manuscript.
We revised the manuscript (red color) and have provided a point-by-point response to your comments below.
Reviewer 1
Although this is a good research paper on Clinical prognosis of lung cancer patients with chronic kidney disease, there are several major points that should be addressed:
C1. The authors claim that they used Chest computed tomography (CT), however there are no data in the results. The authors should add these CT images in order to evaluate the treatment response.
A1. Thank you for your comment. I agree that the sentence in the main text may confuse the reader. (“Chest computed tomography (CT) was performed after every 1–3 months to evaluate the treatment response.”) This study was a retrospective study, so it was mean that the clinical records of the clinicians who treated lung cancer evaluating the treatment response using the conventional CT method were analyzed. Therefore, it has been rewritten as follows (pages 7, lines 6-7):
“The evaluation of the treatment response for lung cancer was confirmed by the clinician's medical record based on the chest CT results performed every 1-3 months.”
C2. The authors should also analyze CKD prevalence and absolute number of CKD patients in different cancer types. They should also perform Multiple logistic regression of various cancer types for CKD.
A2. Thanks for your advice. We fully agree with your opinion. Depending on the type of cancer, the effect of renal dysfunction on the prognosis of cancer may be different. Na et al. [1] compared the prevalence and prognosis of various cancer types (kidney and urinary tract cancer, hematologic malignancy, and other solid cancers [stomach, lung, liver, colorectal, breast, cervix, thyroid, other sites]) according to the presence or absence of CKD. Interestingly, the presence of CKD was an independent poor prognostic factor in cancer-specific mortality except for lung cancer. This result is similar to previously published data [2]. They reported that patients with CKD and lung cancer have a similar clinical course and survival as compared to those with lung cancer and without CKD because of relative rapid progression. Therefore, in the introduction (or discussion) section of the revised manuscript, the prevalence and prognosis of various cancer types with and without CKD analyzed in previous studies were cited (page 14-15, lines 17-23):
“Lung cancer appears to be less affected by the presence or absence of CKD than other types of cancer. Weng et al.[1] studied whether CKD acts as a poor prognostic factor in various types of cancer. As a result, the presence of CKD adversely affected liver cancer, kidney and urinary tract cancer, but not lung cancer. Na et al. [2] also compared the prevalence and prognosis of various cancer types according to the presence or absence of CKD, the presence of CKD was an independent poor prognostic factor in cancer-specific mortality except for lung cancer. They explained that due to the relatively rapid progression of lung cancer, it has shown similar clinical course and survival compared to lung cancer patients and those without CKD.”
Reference
[1] Weng, Pei-Hsuan, et al. "Cancer-specific mortality in chronic kidney disease: longitudinal follow-up of a large cohort." Clinical journal of the American Society of Nephrology 6.5 (2011): 1121-1128.
[2] Na, Sun Young, et al. "Chronic kidney disease in cancer patients: an independent predictor of cancer-specific mortality." American journal of nephrology 33.2 (2011): 121-130.
C3. Circulating blood counts of ALC, AEC1, AMC, Platelet count, NLR1 PLR1 etc should be included in the results analysis
A3. Thank you for your advice. All the substances you mentioned have been found to be prognostic factors for lung cancer. Please understand that acetyl-l-carnitine (ALC) and alveolar epithelial cells (AEC) could not be further analyzed due to the limitations of our study as a retrospective study. However, the absolute monocyte count (AMC), neutrophil-lymphocyte ratio (NLR) and platelet-lymphocyte ratio (PLR) were available additional information from our data. Therefore, the results of analysis of the updated data with AMC, NLR, and PLR were reflected Table 1 in the revised manuscript.
C4. Clinical analysis of patients should include a detailed cytokine analysis of TNF-a, IL-6, IL8, IP-10, IL-6, CXCL5, CXCL9, and CXCL10, CXCL9, CXCL10, CXCL11, CXCL19, etc. The authors should also compare cytokine levels before and after treatment.
A4. Thank you for your advice. Since all the factors mentioned are cytokines related to the prognosis of lung cancer, comparing the levels before and after lung cancer treatment will yield clinically important results. Unfortunately, our study is a retrospective study and has limitations in analyzing the effects of cytokines in patients with lung cancer. However, it would be interesting to analyze the relationship between cytokines and lung cancer co-existing CKD in future studies.
C5. The authors have no data of Survival according to CKD during chemotherapy. They should add Survival data on Time to treatment failure (TTF), progression-free survival (PFS), and overall survival (OS) of patients with CKD during chemotherapy.
A5. Thanks for pointing out this important issue. we analyzed competing risks survival analysis according to other treatment methods (surgery, palliative treatment) as well as chemotherapy, but there was no statistical significance due to the small size of the subgroup. The following shows the results of our analysis on patients who underwent surgery. According to your recommendation, we provide detailed clinical outcomes (overall survival and progression-free survival) according to treatment options including chemotherapy in Supplementary Table 2.
C6. The authors should also add Kaplan–Meier survival curve for lung cancer patients with and without chronic kidney disease (CKD) for comparison.
A6. Thank you for your comment. The results of the impact on the prognosis of lung cancer according to the presence or absence of CKD were described in the revised manuscript by reviewing previous studies. This is described in the introduction section (page5, lines 8-14) and discussion section (page 14-15, lines 17-23, 1-3).
“CKD is associated with poor prognosis in various type of cancer compared to non-CKD patients [7,8]. The impact of CKD on the prognosis of lung cancer patients remains controversial. Several studies reported that the presence or absence of CKD in lung cancer patients does not affect the prognosis of lung cancer [9,10]. By contrast, in a large population-based cohort study, Wei et al. [11] found an increased mortality rate in lung cancer patients with CKD compared to those without CKD. Therefore, the outcome of lung cancer patients with CKD remains unclear.”
“Lung cancer appears to be less affected by the presence or absence of CKD than other types of cancer. Weng et al.[7] studied whether CKD acts as a poor prognostic factor in various types of cancer. As a result, the presence of CKD adversely affected liver cancer, kidney and urinary tract cancer, but not lung cancer. Na et al. [8] also compared the prevalence and prognosis of various cancer types according to the presence or absence of CKD, the presence of CKD was an independent poor prognostic factor in cancer-specific mortality except for lung cancer. They explained that due to the relatively rapid progression of lung cancer, it has shown similar clinical course and survival compared to lung cancer patients and those without CKD. In a study conducted by Lu et al.[10], lung cancer patients with moderate CKD, there is no effect on mortality according to lung cancer stage or CKD stage.”
[References]
- Weng, P.-H.; Hung, K.-Y.; Huang, H.-L.; Chen, J.-H.; Sung, P.-K.; Huang, K.-C. Cancer-specific mortality in chronic kidney disease: longitudinal follow-up of a large cohort. Clinical journal of the American Society of Nephrology 2011, 6, 1121-1128.
- Na, S.Y.; Sung, J.Y.; Chang, J.H.; Kim, S.; Lee, H.H.; Park, Y.H.; Chung, W.; Oh, K.-H.; Jung, J.Y. Chronic kidney disease in cancer patients: an independent predictor of cancer-specific mortality. American journal of nephrology 2011, 33, 121-130.
- Patel, P.; Henry, L.L.; Ganti, A.K.; Potti, A. Clinical course of lung cancer in patients with chronic kidney disease. Lung cancer 2004, 43, 297-300.
- Lu, M.S.; Chen, M.F.; Lin, C.C.; Tseng, Y.H.; Huang, Y.K.; Liu, H.P.; Tsai, Y.H. Is chronic kidney disease an adverse factor in lung cancer clinical outcome? A propensity score matching study. Thoracic Cancer 2017, 8, 106-113.
- Wei, Y.-F.; Chen, J.-Y.; Lee, H.-S.; Wu, J.-T.; Hsu, C.-K.; Hsu, Y.-C. Association of chronic kidney disease with mortality risk in patients with lung cancer: a nationwide Taiwan population-based cohort study. BMJ open 2018, 8, e019661.
C7. The authors do not display any data of Cancer stage survival according to the presence or not of chronic kidney disease and stage of chronic kidney disease. It is really important in order to verify their results
A7. Thanks for pointing out this important issue. We re-evaluated whether renal function really played a role in the prognosis of lung cancer patients, and whether the aggressiveness of lung cancer itself was more important for the prognosis. We divided the NSCLC subgroup into early stage (I-IIIA) and advanced stage (IIIB-IV) and performed competing risks survival analysis. As a result, the degree of renal impairment influenced mortality but not on the progression of early NSCLC (I-IIIA). However, CKD stage did not affect both progression and survival in advanced NSCLC (IIIB-IV). In other words, although renal function affects the clinical course of early NSCLC, the aggressiveness of lung cancer itself seems to more impact on the prognosis of advanced NSCLC. In the revised manuscript, the effect of CKD stage according to lung cancer stage was evaluated using competing risks survival analysis and described in figure 2 and text (page9-10, lines19-22, 1-2).
“To evaluate the effect of competitive risk factors on mortality from lung cancer, a subgroup analysis was performed in patients with NSCLC. The results of competing risks survival analysis according to CKD stage in early NSCLC (stage I–IIIA, Figure 2A) and advanced NSCLC (stage IIIB–IV, Figure 2B) are presented. The effect of CKD stage on death in early NSCLC (stage ≤ IIIA) was significantly highest in the order of CKD4, CKD 5 (hemodialysis), and CKD 3. (Gray’s test, p < 0.001).”
C8. The authors should add a distribution analysis of Charlson comorbidities index (CCI) in CKD and non-CKD lung cancer patients
C9. Thank you for pointing this out. For patients with lung cancer co-existing moderate CKD, it is important to review other comorbidities. We investigated the CCI of all patients enrolled in our study and updated all statistical results. As a result, the median CCI was 10.0 (8.0-11.0), and in multivariate analysis, when CCI ≥ 10, the risk of mortality increased approximately 1.7 times higher. The revised results related to CCI are described in Tables 1, 3, and in the Result section (pages 10-11, lines 17-21, 1-3) and the Discussion section (page 12, lines 13-20) as follows.
“From the univariate analysis, old age (≥75 years), a low BMI (≤23 kg/m2), Charlson comorbidities index (CCI ≥ 10), comorbid chronic airway disease, SCLC, stage IV NSCLC, absolute monocyte count (AMC ≥ 500, /μL), and neutrophil-lymphocyte ratio (NLR ≥ 3.37) were associated with poor prognosis in terms of reduced OS (all p < 0.05). From the multivariate analysis, age ≥75 years (aHR, 1.658; 95% CI, 1.148–2.394, p = 0.007), CCI ≥ 10 (aHR, 1.669; 95% CI, 1.069–2.605, p = 0.024) and stage IV NSCLC (aHR, 2.395; 95% CI, 1.512–3.796, p <0.001) were independent prognostic factors.”
“The median CCI value of the subjects in this study was 10.0 (8.0-11.0), and patients with CCI higher than this had an approximately 1.7-times higher risk of mortality. Marcus et al. [20] shown that higher CCI was associated with higher lung cancer-specific mortality (CCI ≥ 3: aHR, 5.16; 95% CI, 2.07–12.89). Lu et al. [10] reported that median CCI was 8.0 in moderate CKD patients with lung cancer, and median survival for patients with a CCI > 9 was 6 months compared with 10 months in patients with CCI < 9. Comorbidities have a significant impact on the survival of lung cancer patients, and the number and severity of comorbidities are likely to influence treatment options for these patients.”
Overall this paper is presented with low scientific novelty and originality and cannot be accepted in its current form. Major revision should be performed.
I fully agree with your opinion and have made as modifications as possible according to your recommendations. Lung cancer with moderate CKD is so rare that it is difficult for clinicians to gain treatment experience. Therefore, I believe that it can be helpful to clinicians to show the clinical features, treatment options, and prognosis of this rare lung cancer with CKD in various studies. Furthermore, by analyzing the prognosis according to the stage of lung cancer with CKD with your advice, we were able to descript a consideration to "in the early stage of NSCLC, mortality rate was affected according to the CKD stage, but not in the advanced stage NSCLC." In these respects, we look forward to the opportunity of this study to be helpful to other researchers and clinicians.

Reviewer 2 Report
Dear Authors
Congratulations on your interesting analysis of a difficult problem. I have read your work with interest, I have some comments / questions. 1. Was the time of lowering eGFR been verified in the definition of CKD used in the study? You described that CKD was diagnosed at the time of cancer diagnosis. By definition, however, it is necessary to confirm persistent renal dysfunction (or morphological changes) for more than 3 months. I believe that you must refer to this issue in the text. 2. I suggest refining the graphic presentation of the text - the data provided in the tables are very accurate, but difficult for the reader to analyze. Perhaps you will be able to replace some tabular data with charts? for example, the results of a multivariate regression analysis? 3. I have the impression that the hypothesis explaining the relationship between the degree of CKD and prognosis is not sufficiently emphasized in the discussion. Do you not think that it is the initially bad eGFR that determines the treatment used and that it affects the prognosis? Have you analyzed how tDear Authors Congratulations on your interesting analysis of a difficult problem. I have read your work with interest, I have some comments / questions. 1. Was the time of lowering eGFR been verified in the definition of CKD used in the study? You described that CKD was diagnosed at the time of cancer diagnosis. By definition, however, it is necessary to confirm persistent renal dysfunction (or morphological changes) for more than 3 months. I believe that you must refer to this issue in the text. 2. I suggest refining the graphic presentation of the text - the data provided in the tables are very accurate, but difficult for the reader to analyze. Perhaps you will be able to replace some tabular data with charts? for example, the results of a multivariate regression analysis? 3. I have the impression that the hypothesis explaining the relationship between the degree of CKD and prognosis is not sufficiently emphasized in the discussion. Do you not think that it is the initially bad eGFR that determines the treatment used and that it affects the prognosis? Have you analyzed how the treatment you take affects the prognosis?he treatment you take affects the prognosis?Author Response
Point-by-point response to the reviewers’ comments
Manuscript title: Clinical prognosis of lung cancer in patients with moderate chronic kidney disease
To the reviewers
We sincerely appreciate the time and effort you invested to help us improve this manuscript.
We revised the manuscript (red color) and have provided a point-by-point response to your comments below.
Reviewer 2
Congratulations on your interesting analysis of a difficult problem. I have read your work with interest, I have some comments / questions.
C1. Was the time of lowering eGFR been verified in the definition of CKD used in the study? You described that CKD was diagnosed at the time of cancer diagnosis. By definition, however, it is necessary to confirm persistent renal dysfunction (or morphological changes) for more than 3 months. I believe that you must refer to this issue in the text.
C1. Thank you for your comment. As you mentioned, the period of decreased renal function is very important in the diagnosis of CKD. Therefore, in the revised manuscript, the definition of CKD has been modified as follows (page6, lines 15-17):
“CKD was defined as an eGFR <90 mL/min/1.73 m2 with 1) a history of CKD, 2) follow-up at the hospital for more than 3 months, or 3) morphological change was confirmed at the time of lung cancer diagnosis.
C2. I suggest refining the graphic presentation of the text - the data provided in the tables are very accurate, but difficult for the reader to analyze. Perhaps you will be able to replace some tabular data with charts? for example, the results of a multivariate regression analysis?
C3. I have the impression that the hypothesis explaining the relationship between the degree of CKD and prognosis is not sufficiently emphasized in the discussion. Do you not think that it is the initially bad eGFR that determines the treatment used and that it affects the prognosis? Have you analyzed how the treatment you take affects the prognosis?
A3. Thank you for pointing this out. In previous studies, lung cancer appears to be less affected by the presence or absence of CKD than other types of cancer. Weng et al.[1] studied whether CKD acts as a poor prognostic factor in various types of cancer. As a result, the presence of CKD adversely affected liver cancer, kidney and urinary tract cancer, but not lung cancer. Na et al. [2] also compared the prevalence and prognosis of various cancer types according to the presence or absence of CKD, the presence of CKD was an independent poor prognostic factor in cancer-specific mortality except for lung cancer. They explained that due to the relatively rapid progression of lung cancer, it has shown similar clinical course and survival compared to lung cancer patients than those without CKD. In a study conducted by Lu et al.[3], lung cancer patients with moderate CKD, there is no effect on mortality according to lung cancer stage or CKD stage.
Therefore, we re-evaluated whether renal function really played a role in the prognosis of lung cancer patients, and whether the aggressiveness of lung cancer itself was more important for the prognosis. We divided the NSCLC subgroup into early stage (I-IIIA) and advanced stage (IIIB-IV) and performed competing risks survival analysis. As a result, the degree of renal impairment influenced mortality but not on the progression of early NSCLC (I-IIIA). However, CKD stage did not affect both progression and survival in advanced NSCLC (IIIB-IV). In other words, although renal function affects the clinical course of early NSCLC, the aggressiveness of lung cancer itself seems to more impact on the prognosis of advanced NSCLC. In the revised manuscript, the effect of CKD stage according to lung cancer stage was evaluated using competing risks survival analysis and described in figure 2 and text (page9-10, lines19-22, 1-2).
“To evaluate the effect of competitive risk factors on mortality from lung cancer, a subgroup analysis was performed in patients with NSCLC. The results of competing risks survival analysis according to CKD stage in early NSCLC (stage I–IIIA, Figure 2A) and advanced NSCLC (stage IIIB–IV, Figure 2B) are presented. The effect of CKD stage on death in early NSCLC (stage ≤ IIIA) was significantly highest in the order of CKD4, CKD 5 (hemodialysis), and CKD 3. (Gray’s test, p < 0.001).”
[References]
- Weng, P.-H.; Hung, K.-Y.; Huang, H.-L.; Chen, J.-H.; Sung, P.-K.; Huang, K.-C. Cancer-specific mortality in chronic kidney disease: longitudinal follow-up of a large cohort. Clinical journal of the American Society of Nephrology 2011, 6, 1121-1128.
- Na, S.Y.; Sung, J.Y.; Chang, J.H.; Kim, S.; Lee, H.H.; Park, Y.H.; Chung, W.; Oh, K.-H.; Jung, J.Y. Chronic kidney disease in cancer patients: an independent predictor of cancer-specific mortality. American journal of nephrology 2011, 33, 121-130.
- Lu, M.S.; Chen, M.F.; Lin, C.C.; Tseng, Y.H.; Huang, Y.K.; Liu, H.P.; Tsai, Y.H. Is chronic kidney disease an adverse factor in lung cancer clinical outcome? A propensity score matching study. Thoracic Cancer 2017, 8, 106-113.

Round 2
Reviewer 1 Report
Accept after minor revision (corrections to minor methodological errors and text editing)
Reviewer 2 Report
Dear Authores
Thank you for concidering my comments and suggestions. I am fully satisfied with the modifications introduced and the explanations provided
I support the publication of your article
Best regards